# DisCo: Disentangled Control for Realistic Human Dance Generation

## Abstract

Generative AI has made significant strides in computer vision, particularly in text-driven image/video synthesis (T2I/T2V). Despite the notable advancements, it remains challenging in human-centric content synthesis such as realistic dance generation. Current methodologies, primarily tailored for human motion transfer, encounter difficulties when confronted with real-world dance scenarios (*e.g.*, social media dance) which require to generalize across a wide spectrum of poses and intricate human details. In this paper, we depart from the traditional paradigm of human motion transfer and emphasize two additional critical attributes for the synthesis of human dance content in social media contexts: *(i) Generalizability:* the model should be able to generalize beyond generic human viewpoints as well as unseen human subjects, backgrounds, and poses; *(ii) Compositionality:* it should allow for composition of seen/unseen subjects, backgrounds, and poses from different sources seamlessly. To address these challenges, we introduce DisCo, which includes a novel model architecture with disentangled control to improve the compositionality of dance synthesis, and an effective human attribute pre-training for better generalizability to unseen humans. Extensive qualitative and quantitative results demonstrate that DisCo can generate high-quality human dance images and videos with diverse appearances and flexible motions.

## 1 Introduction

Starting from the era of GAN (Brock et al., 2018; Goodfellow et al., 2020), researchers (Siarohin et al., 2019b; 2021) try to explore the human motion transfer by transferring talking and Tai-Chi poses from a source image to a target individual. It requires the generated images/videos to precisely follow the source pose and retain the appearance of human subjects and backgrounds from the target image. However, when it comes to much more diverse and nuanced visual contents such as TikTok dancing videos, GANs tend to struggle due to mode collapse, capturing only a limited portion of the real data distribution (Figure 1a, MRAA (Siarohin et al., 2021)).

Recently, diffusion-based generative models (Ho et al., 2020; Song et al., 2020a;b) have significantly improved the synthesis in both diversity and stability. The introduction of ControlNet (Zhang & Agrawala, 2023) further enhances the controllability by injecting geometric conditions (*e.g.*, human skeleton) into Stable Diffusion (SD) model (Rombach et al., 2022), thus becomes possible to be utilized in human dance generation. However, prevailing ControlNet-based methods either rely on guidance from coarse-grained text prompts (Figure 1a, T2I Adaptor (Mou et al., 2023)) or simply substitute the text condition in T2I/T2V models with the referring image (Figure 1a, DreamPose (Karras et al., 2023)). It remains unclear how to ensure the consistency of rich human semantics and background in real-world dance scenarios. Moreover, almost all existing methods are trained on insufficient dance video datasets, hence suffer from either limited human attributes (Wang et al., 2018; Chan et al., 2019; Zhou et al., 2019) or excessively simple poses and scenes (Li et al., 2021; Liu et al., 2019), leading to the poor zero-shot generalizability to unseen human dance scenarios.

In order to support real-life applications, such as user-specific short video generation, we step from the conventional human motion transfer and further highlight two properties in human dance synthesis:

- *Generalizability*: The model should be able to generalize to hard cases, *e.g.*, non-generic human view as well as unseen human subject, background and pose (Figure 1a).

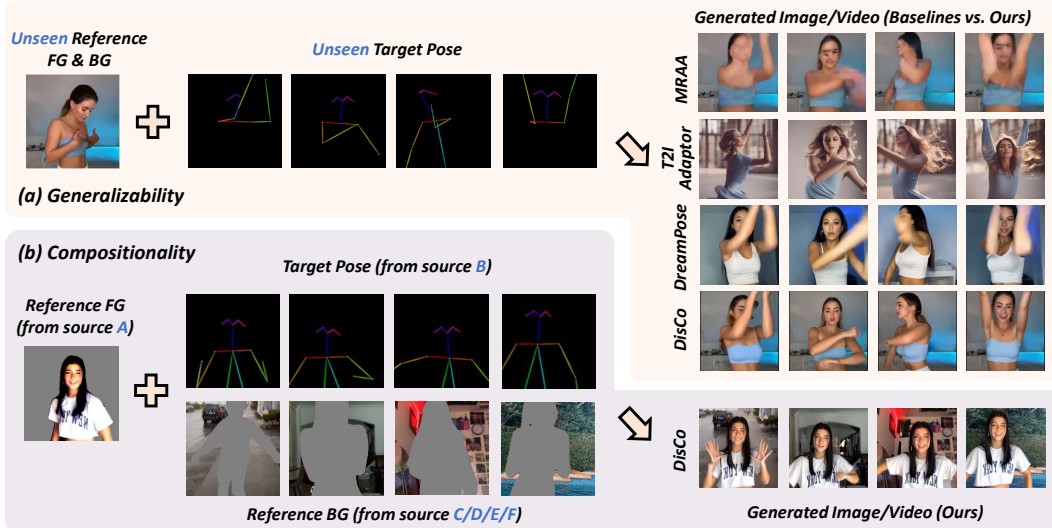

Figure 1: We propose DISCO for human dance generation on social media platforms, focusing on two key properties compared to conventional human motion transfer: (a) **Generalizability**: generalizable to unseen human subject (FG), background (BG) and pose; (b) **Compositionality**: adapting to the arbitrary composition of FG, BG and pose, each from a different source.

- *Compositionality*: The generated images/videos can be from an arbitrary composition of seen or unseen human subject, background and pose, sourced from different images/videos (Figure 1b).

In this regard, we propose a novel approach, DISCO, for realistic human dance generation in social media. DISCO consists of two key designs: (*i*) a novel *model architecture with disentangled control* for improved faithfulness and compositionality; and (*ii*) an effective pre-training strategy with DISCO for better generalizability, named *human attribute pre-training*.

**Model Architecture with Disentangled Control** (Section 3.2). We attribute the failure of existing ControlNet-based methods to the inappropriate integration of various conditions. In this paper, we apply ControlNet for background and human pose. Differently, we utilize the VAE as the background encoder to fully leverage the prior of semantically rich images, while a tiny convolutional encoder for highly abstract skeleton. For the human subject, we incorporate its CLIP image embedding with the denoising U-Net as well as all other conditions via the cross-attention modules, to help dynamic foreground synthesis. By disentangling the control from three conditions, DISCO can not only enable arbitrary compositionality of human subjects, backgrounds, and dance-moves (Figure 1b), but also achieve high fidelity via the thorough utilization of the various input conditions (check Table 4 & 5 for the ablation of the condition mechanism).

**Human Attribute Pre-training** (Section 3.3). We design a novel proxy task in which the model conditions on the separate foreground and background region features and must reconstruct the complete image. This task is non-trivial as it enables the model to (*a*) effectively distinguish the dynamic human subject and static background for the ease of following pose transfer; (*b*) better encode-and-decode the complicated human faces and clothes during pre-training, and leaves the pose control learning to the fine-tuning stage of human dance synthesis. Crucially, without the constraint of pairwise human images for pose control, we can overcome the insufficiency of high-quality dance video data by leveraging large-scale collections of human images to learn diverse human attributes, in turn, greatly improve the generalizability of DISCO to unseen humans.

Our contributions are summarized as three-folds:

- We highlight two properties, generalizability and compositionality, that are missing from the conventional human motion transfer for more challenging social media dance synthesis problem, to facilitate its potential in the production of user-specific short videos.

- To address this problem, we propose DISCO framework with (*i*) a novel model architecture for disentangled control to ensure compositionality in generation; and (*ii*) an effective human attribute pre-training to improve generalizability to unseen humans and non-generic views.

- We conduct a broad variety of evaluations and applications to demonstrate the effectiveness of DISCO. Notably, even without temporal consistency modeling, DISCO can already achieve superior FID (**28.31** v.s 53.78) and FID-VID (**55.17** v.s 66.36) scores over the state-of-the-art approaches. Adding temporal modeling further boosts FID-VID scores of DISCO to **29.37**.

## 2 RELATED WORK

**Diffusion Models for Controllable Image/Video Generation.** Diffusion probabilistic models (Sohl-Dickstein et al., 2015; Dhariwal & Nichol, 2021) have shown great success in high-quality image/video generation. Towards user-specific generation, text prompts are first utilized as the condition for image generation (Ramesh et al., 2022; Ho & Salimans, 2022; Saharia et al., 2022; Xu et al., 2022). Among them, Stable Diffusion (Rombach et al., 2022) (SD) stands as the representative work to date, with high efficiency and competitive quality via diffusion over the latent space. For better controllability, ControlNet (Zhang & Agrawala, 2023) introduces additional control mechanisms into SD beyond texts, such as sketch, human skeleton, and segmentation map. Compared to image, text-to-video synthesis (Ho et al., 2022; Singer et al., 2022; Esser et al., 2023; Wu et al., 2022; Khachatryan et al., 2023), is more challenging due to the lack of well-annotated data and difficulties in temporal modeling. Thus, existing controllable video generation methods (Ma et al., 2023; Khachatryan et al., 2023) mainly stem from pre-trained text-to-image model and try to introduce motion/pose prior to the text-to-video synthesis. In this work, we look into a more challenging setting of conditional human image/video synthesis which requires precise control of both human attributes (such as identity, clothing, makeup, hairstyle, *etc.*) as well as the dance-moves (poses) in social media dance scenarios.

**Human Dance Synthesis.** Early work on this task includes video-to-video synthesis (Wang et al., 2018; 2019; Gafni et al., 2019; Chan et al., 2019), still image animation (Weng et al., 2019; Holynski et al., 2021; Wang et al., 2022; Blattmann et al., 2021; Yoon et al., 2021) and motion transfer (Zhou et al., 2019; Siarohin et al., 2019a; Lee et al., 2019; Suo et al., 2022). Nevertheless, these methods require either a several-minute-long target person video for human-specific fine-tuning, or multiple separate networks and cascaded training stages for background, motion and occlusion map prediction. The advances of diffusion models (Rombach et al., 2022) greatly simplify the training of such generative models, inspiring follow-up diffusion models (Kumari et al., 2022; Ni et al., 2023) tailored for human dance generation. Still, these methods require a separate motion prediction module and struggle to precisely control the human pose. DreamPose (Karras et al., 2023) is perhaps the most relevant study to ours, which proposes an image-and-pose conditioned diffusion method for still fashion image animation. However, as they consider only fashion subjects with easy catwalk poses in front of an empty background, their model may suffer from limited generalization ability, prohibiting its potential for more intricate human dance synthesis in real-world scenarios.

## 3 DISCO

We start by first formally introducing the setting for social media dance generation. Let $f$ and $g$ represent human foreground and background in the reference image. Given a specific (or a sequence of) pose keypoint $p = p_t$ (or $p = \{p_1, p_2, ..., p_T\}$), we aim to generate **realistic** images $I_t$ (or videos $V = \{I_1, I_2, ..., I_T\}$) conditioned on $f, g, p$. The generated images (or videos) should be 1) *faithful*: the human attribute and background of the synthesis should be consistent with $f$ and $g$ from the reference image and the generated human subject should be aligned with the pose $p$; 2) *generalizable*: the model should be able to generalize to unseen humans, backgrounds and poses, without the need of human-specific fine-tuning; and 3) *composable*: the model should adapt to arbitrary composition of $f, g, p$ from different image/video sources to generate novel images/videos. In what follows, Section 3.1 briefly reviews the latent diffusion models and ControlNet, which are the basis of DISCO. Section 3.2 details the model architecture of DISCO with disentangled control of human foreground, background, and pose to enable faithful and fully composable human dance image/video synthesis. Section 3.3 presents how to further enhance the generalizability of DISCO, as well as the faithfulness in generated contents by pre-training human attributes from large-scale human images. The overview of DISCO can be found in Figure 2.

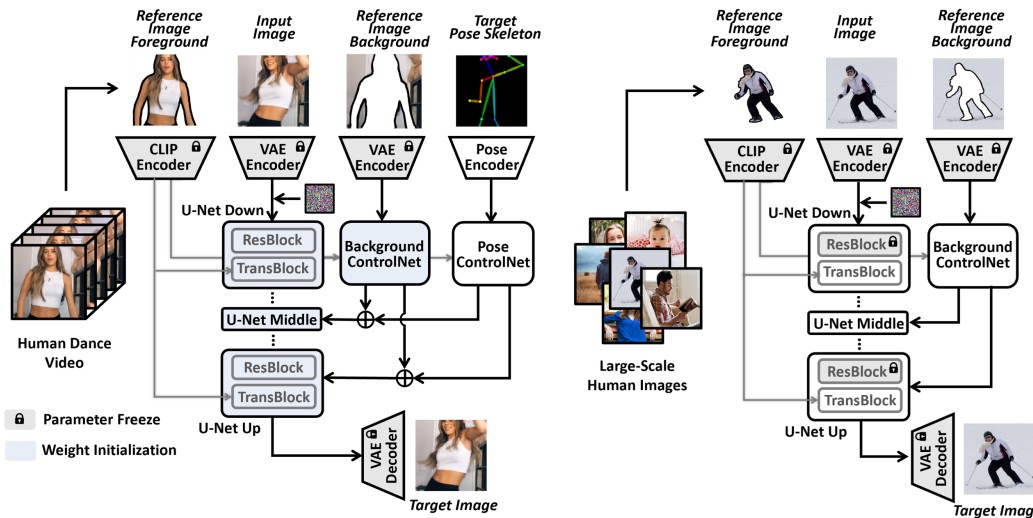

(a) Model Architecture with Disentangled Control      (b) Human Attribute Pre-training

Figure 2: The proposed DISCO framework for social media human dance generation.

## 3.1 PRELIMINARY: LATENT DIFFUSION MODELS & CONTROLNET

**Latent Diffusion Models** (LDM) is a type of diffusion model that operates in the encoded latent space of an autoencoder $\mathcal{D}(\mathcal{E}(\cdot))$. An exemplary LDM is the popular Stable Diffusion (SD) (Rombach et al., 2022) which consists an autoencoder VQ-VAE (Van Den Oord et al., 2017) and a time-conditioned U-Net (Ronneberger et al., 2015) for noise estimation. A CLIP ViT-L/14 text encoder (Radford et al., 2021) is used to project the input text query into the text embedding condition $c_{\text{text}}$.

During training, given an image $I$ and the text condition $c_{\text{text}}$, the image latent $z_0 = \mathcal{E}(I)$ is diffused in $T$ time steps with a deterministic Gaussian process to produce the noisy latent $z_T \sim \mathcal{N}(0, 1)$. SD is trained to learn the reverse denoising process with the following objective (Rombach et al., 2022):

$$L = \mathbb{E}_{\mathcal{E}(I), c_{\text{text}}, \epsilon \sim \mathcal{N}(0,1), t} \left[ \|\epsilon - \epsilon_\theta(z_t, t, c_{\text{text}})\|_2^2 \right], t = 1, ..., T$$

where $\epsilon_\theta$ represents the trainable modules, containing a U-Net architecture composed of the convolution (ResBlock) and self-/cross-attention (TransBlock), which accepts the noisy latents $z_t$ and the text embedding condition $c_{\text{text}}$ as the input. After training, one can apply a deterministic sampling process (*e.g.*, DDIM (Song et al., 2020a)) to generate $z_0$ and pass it to the decoder $\mathcal{D}$ towards the final image.

**ControlNet** (Zhang & Agrawala, 2023), built upon SD, manipulates the input to the intermediate layers of the U-Net in SD, for controllable image generation. Specifically, it creates a trainable copy of the U-Net down/middle blocks and adds an additional "zero convolution" layer. The outputs of each copy block is then added to the skip connections of the original U-Net. Apart from the text condition $c_{\text{text}}$, ControlNet is trained with an additional external condition vector $c$ which can be many types of condition, such as edge map, depth map and segmentation.

## 3.2 MODEL ARCHITECTURE WITH DISENTANGLED CONTROL

The direct application of ControlNet to social media dance generation presents challenges due to the missing of reference human image condition, which is critical for keeping the human identity and attribute consistent in the synthesized images/videos. Recent explorations in image variations (Justin & Lambda, 2022) replace the CLIP text embedding with the CLIP image embedding as the SD condition, which can retain some high-level semantics from the reference image. Nevertheless, the geometric/structural control onto the generated image is still missing.

Taking the distinctive benefits of these two different control designs, we introduce a novel model architecture with disentangled control, to enable accurate alterations to the human pose, while simultaneously maintaining attribute and background stability. Meanwhile, it also facilitates full compositionality in the human dance synthesis, accommodating any combination of human foreground, pose, and background (Figure 1b). Specifically, given a reference human image, we can first utilize

an existing human matting method (*e.g.*, SAM (Kirillov et al., 2023; Liu et al., 2023b)) to separate the human foreground from the background. Next, we explain how all three conditions, the human foreground $f$, the background $g$ and the desired pose $p$, are incorporated into DISCO.

**Referring Foreground via Cross Attention**. To help model easily adapt to the CLIP image feature space, we first use the pre-trained image variation latent diffusion model (Justin & Lambda, 2022) for the U-Net parameter initialization. However, in contrast to using the global CLIP image embeddings employed by image variation methods, here we adopt the local CLIP image embeddings right before the global pooling layer, for more fine-grained human semantics encoding. Consequently, the original text embedding $c_{\text{text}} \in \mathbb{R}^{l \times d}$ is superseded by the local CLIP image embeddings of the human foreground $c_f \in \mathbb{R}^{hw \times d}$ to serve as the key and value feature in cross-attention layer, where $l, h, w, d$ represent the caption length, the height, width of the visual feature map and the feature dimension.

**Controlling Background and Pose via ControlNets**. For pose $p$, we adopt the vanilla design of ControlNet. Specifically, we embed the pose image into the same latent space as the Unet input via four convolution layers, and dedicate a ControlNet branch $\tau_\theta$ to learn the pose control. For background $g$, we insert another ControlNet branch $\mu_\theta$ to the model. Notably, we propose to use the pre-trained VQ-VAE encoder $\mathcal{E}$ in SD, instead of four randomly initialized convolution layers, to convert the background image into dense feature maps to preserve intricate details. The remainder of the architecture for the background ContorlNet branch follows the original ControlNet. As we replace the text condition with the referring foreground in the cross-attention modules, we also update the condition input to ControlNet as the local CLIP image feature of the referring foreground. As shown in Figure 2a, the outputs of the two ControlNet branches are combined via addition and fed into the middle and up block of the U-Net.

With the design of the disentangled controls above, we fine-tune DISCO with the same latent diffusion modeling objective (Rombach et al., 2022):

$$L = \mathbb{E}_{\mathcal{E}(I), c_f, \tau_\theta(p), \mu_\theta(g), \epsilon \sim \mathcal{N}(0,1), t} \left[ \| \epsilon - \epsilon_\theta(z_t, t, c_f, \tau_\theta(p), \mu_\theta(g)) \|_2^2 \right],$$

where $\epsilon_\theta$, $\tau_\theta$ and $\mu_\theta$ are the trainable network modules. Specifically, $\epsilon_\theta$ contains the U-Net architecture composed of the convolution (ResBlock) and self-/cross-attention (TransBlock), which accepts the noisy latents $z_t$ and the referring foreground condition $c_f$ as the inputs. $\tau_\theta$ and $\mu_\theta$ represent the two ControlNet branches for pose condition $p$ and background condition $g$, respectively.

**Temporal Modeling (TM).** To achieve better temporal continuity for video output, we follow (Singer et al., 2022; Esser et al., 2023) to introduce a 1D temporal convolution/attention layer after the existing 2D spatial ones in ResBlock and TransBlock (Figure 3). Besides, the temporal layers are zero-initialized, and connected via residual connections. Simultaneously, we adjust the model input from image to video by concatenating the pose $p$ from $n$ consecutive frames, duplicating the background $g$ and the initial noise for $n$ times.

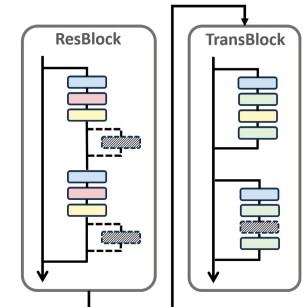

Figure 3: The detailed architecture of the ResBlock and TransBlock. The temporal convolution/attention module (dotted box) is optional.

### 3.3 HUMAN ATTRIBUTE PRE-TRAINING

In utilizing the disentangled control architecture for DISCO, although it shows promises in pose control and background reconstruction, we find it remains challenging to have faithful generations with unseen human subject foregrounds and non-generic human views, demonstrating poor generalizability. The crux of this matter lies in the current training pipeline. It relies on high-quality human videos to provide training pairs of human images, with the same human foreground and background appearance, but different poses. Yet, we observe that current training datasets used for human dance generation confront a dilemma of "mutual exclusivity" — they cannot ensure both diversity in human attributes (such as identity, clothing, makeup, hairstyle, *etc.*) and the complicated poses due to the prohibitive costs of collecting and filtering human videos. As an alternative, human images, which are widely available over the internet, contain diverse human subject foregrounds and backgrounds, despite of the missing paired images with pose alterations.

This motivates us to propose a pre-training task with DISCO, human attribute pre-training, to improve the generalizability and the faithfulness in generation when encountering unseen human subjects.

Rather than directly retrieving and constructing the high-quality human dance video dataset, we explore a much more efficient alternative approach, namely *Human Attribute Pre-training*. Figure 2b shows the details. Specifically, we propose to reconstruct the whole image given the human foreground and background features. In this way, model effectively learns the distinguishment between human subject and foreground, as well as diverse human attributes from large-scale images. Compared to the human dance generation fine-tuning, the ControlNet branch for pose control is removed while the rest of the architecture remains the same. Consequently, we modify the objective as:

$$L = \mathbb{E}_{\mathcal{E}(I), c_f, \mu_\theta(g), \epsilon \sim \mathcal{N}(0,1), t} \left[ \| \epsilon - \epsilon_\theta(z_t, t, c_f, \mu_\theta(g)) \|_2^2 \right].$$

Empirically, we find that freezing the ResNet blocks in U-Net during pre-training can achieve better reconstruction quality of human faces and subtleties.

For human dance generation fine-tuning, we initialize the U-Net and ControlNet branch for background control (highlighted with blue in Figure 2a) by the pre-trained model, and initialize the pose ControlNet branch with the pre-trained U-Net weight following (Zhang & Agrawala, 2023).

## 4 EXPERIMENTS

### 4.1 EXPERIMENTAL SETUP

We train the models on the public TikTok dataset (Jafarian & Park, 2021) for referring human dance generation. TikTok dataset consists of about 350 dance videos (with video length of 10-15 seconds) capturing a single-person dance. For each video, we first extract frames with 30fps, and run Grounded-SAM (Kirillov et al., 2023) and OpenPose (Cao et al., 2017) on each frame to infer the human subject mask for separating the foreground from the background and the pose skeleton. 335 videos are sampled as the training split. To ensure videos from the same person (same identity with same/different appearance) are not present in both training and testing splits, we collect 10 TikTok-style videos depicting different people from the web, as the testing split. We train our model on 8 NVIDIA V100 GPUs for 70K steps with image size $256 \times 256$ and learning rate $2e^{-4}$. During training, we sample the first frame of the video as the reference and all others at 30 fps as targets. For equipping TM, we set $n = 8$, and apply a learning rate of $5e^{-4}$ on the temporal convolution/attention layers. Both reference and target images are randomly cropped at the same position along the height dimension with the aspect ratio of 1, before resized to $256 \times 256$. For evaluation, we apply center cropping instead of random cropping.

For human attribute pre-training, we use a combination of multiple public datasets (TikTok [1] (Jafarian & Park, 2021), COCO (Lin et al., 2014), SHHQ (Fu et al., 2022), DeepFashion2 (Ge et al., 2019), LAION (Schuhmann et al., 2021)). We first run Grounded-SAM (Kirillov et al., 2023) with the prompt of "person" to automatically generate the human foreground mask, and then filter out images without human. This results in over 700K images for pre-training. All pre-training experiments are conducted on 32 NVIDIA V100 GPUs for 25K steps with image size $256 \times 256$ and learning rate $1e^{-3}$. We initialize the U-Net model with the pre-trained weights of Stable Diffusion Image Variations (Justin & Lambda, 2022). The ControlNet branches are initialized with the same weight as the U-Net model, except for the zero-convolution layers, following Zhang & Agrawala (2023). After human attribute pre-training, we initialize the U-Net and ControlNet branch for background control by the pre-trained model, and initialize the pose ControlNet branch with the pre-trained U-Net weight, for human dance generation fine-tuning.

### 4.2 DISCO APPLICATIONS

Benefiting from the strong human synthesis capability powered by the disentangled control design as well as human attribute pre-training, our DISCO provides flexible and fine-grained controlability and generalizability to arbitrary combination of human subject, pose and background. Given three existing images, each with distinct human subject, background and pose, there can be a total of 27 combinations. Here, we showcase 5 representatives scenarios in Figure 4 for **human image editing**: (*i*) *Human Subject / Pose Re-targeting*: the model have been exposed to training instances of the human subject or the ones of the pose, but the specific combinations of both are new to the model; (*ii*)

---

[1]We only use the training split for pre-training to avoid potential data leak.

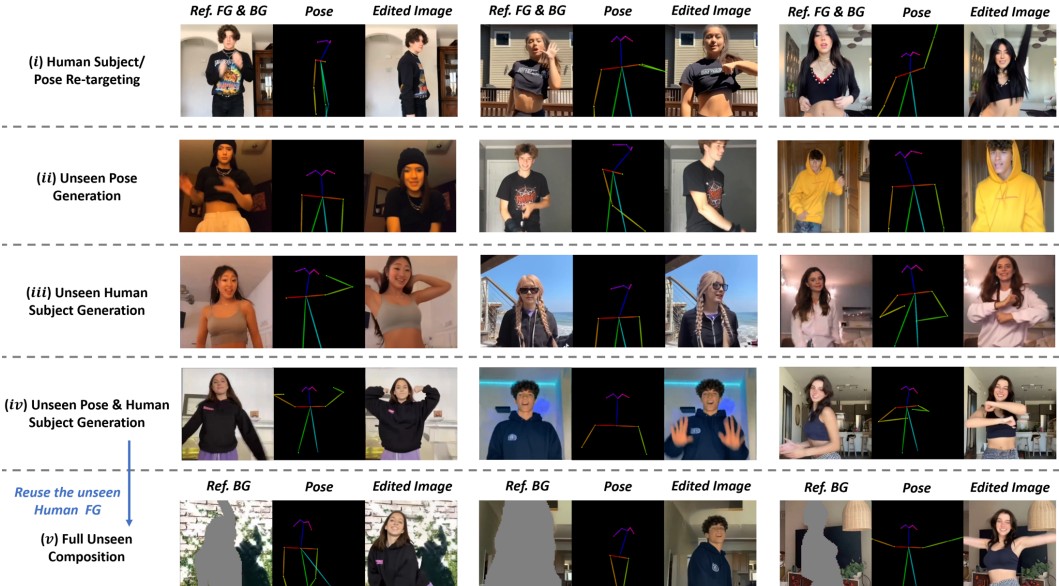

Figure 4: Visualizations of 5 representative scenarios for **human image editing**. DISCO also generalizes to different image ratios and diverse human pose views (*e.g.*, full-body human). Check Figure 10 for details.

*Unseen Pose Generation*: the human subject is from the training set, but the pose is novel, from the testing set; (*iii*) *Unseen Human Subject Generation*: the poses are sampled from the training set, but the human subject is novel, which is either from the testing set or crawled from the web; (*iv*) *Unseen Pose & Human Subject Generation*: both the human subject and pose are not present in the training set; (*v*) *Full Unseen Composition*: sampling a novel background from another unseen image/video based on the "unseen pose & human subject generation". Examples in Figure 4 demonstrate that DISCO can flexibly update one of (or a composition of) human subject/background/pose in a given image to a user-specified one (or composition), either from existing training samples or novel images.

Observing satisfying human image editing results from DISCO, especially the faithfulness in edited images, we can further extend it to **human dance video generation** as it is. Given a reference image and a target pose sequence either extracted from an existing video or from user manipulation of a human skeleton, we generate the video frame-by-frame, with the reference image and a single pose as the inputs to DISCO. We delay the visualization of generated videos and relevant discussions to Figure 5 in the next section. More examples of the two applications above are included in Appendix.

Though it is not the focus of this paper, our final DISCO model can be readily and flexibly integrated with efficient fine-tuning techniques (Hu et al., 2021; Wu et al., 2022; Karras et al., 2023) for subject-specific fine-tuning on one or multiple images of the same human subject. We leave discussions and results of this setting to Appendix B.

## 4.3 MAIN RESULTS

We provide quantitative and qualitative comparisons against both conventional motion transfer methods FOMM (Siarohin et al., 2019b), MRAA (Siarohin et al., 2021), TPS (Zhao & Zhang, 2022), and most relevant work DreamPose (Karras et al., 2023), an image-to-video model designed for fashion domain with densepose control. DreamPose replaces the CLIP text feature with the image embedding with a dual CLIP-VAE encoder and adapter module. Instead of adopting ControlNet, it utilizes a sequence of denseposes (Güler et al., 2018) as the pose condition. It is worth noting that the TikTok dancing videos we evaluate on are all real-world user-generated content, which are much more complicated than those in existing dancing video datasets (Wang et al., 2018; Chan et al., 2019; Jiang et al., 2023), with clean background, same/similar clothing or fixed camera angles.

**Quantitative Comparison**. Since DISCO is applicable to both image and video generation for human dance synthesis, here we compare the models on both image- and video-wise generative metrics. To evaluate the image generation quality, we report frame-wise FID (Heusel et al., 2017), SSIM (Wang et al., 2004), LISPIS (Zhang et al., 2018), PSNR (Hore & Ziou, 2010) and L1; while for videos, we

Table 1: Quantitative comparisons of DISCO with the recent SOTA method DreamPose. "CFG" and "HAP" denote classifier-free guidance and human attribute pre-training, respectively. ↓ indicates the lower the better, and vice versa. For DISCO[†], we further scale up the fine-tuning stage to ~600 TikTok-style videos. Methods with ∗ directly use target image as the input, including more information compared to the OpenPose.

| Method | Image | | | | | Video | |
|---|---|---|---|---|---|---|---|
| | FID ↓ | SSIM ↑ | PSNR ↑ | LISPIS ↓ | L1 ↓ | FID-VID ↓ | FVD ↓ |
| FOMM∗ (Siarohin et al., 2019b) | 85.03 | 0.648 | 29.01 | 0.335 | 3.61E-04 | 90.09 | 405.22 |
| MRAA∗ (Siarohin et al., 2021) | 54.47 | 0.672 | **29.39** | 0.296 | **3.21E-04** | 66.36 | 284.82 |
| TPS∗ (Zhao & Zhang, 2022) | 53.78 | 0.673 | 29.18 | 0.299 | 3.23E-04 | 72.55 | 306.17 |
| DreamPose (Karras et al., 2023) | 79.46 | 0.509 | 28.04 | 0.450 | 6.91E-04 | 80.51 | 551.56 |
| DreamPose (CFG) | 72.62 | 0.511 | 28.11 | 0.442 | 6.88E-04 | 78.77 | 551.02 |
| DISCO (w/o HAP) | 61.06 | 0.631 | 28.78 | 0.317 | 4.46E-04 | 73.29 | 366.39 |
| DISCO (w/. HAP) | 38.19 | 0.663 | 29.33 | 0.291 | 3.69E-04 | 61.88 | 286.91 |
| DISCO (w/. HAP, CFG) | 30.75 | 0.668 | 29.03 | 0.292 | 3.78E-04 | 59.90 | 292.80 |
| DISCO[†] (w/. HAP, CFG) | **28.31** | **0.674** | 29.15 | **0.285** | 3.69E-04 | **55.17** | **267.75** |

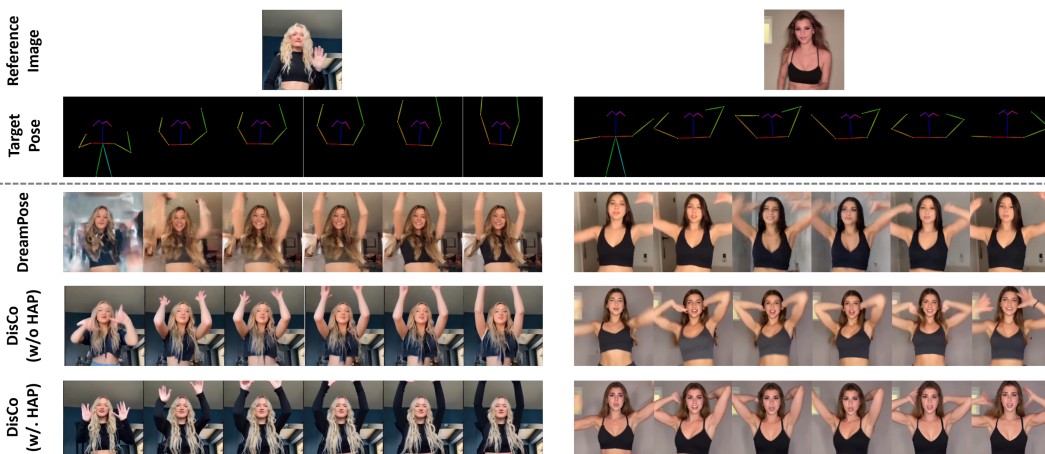

Figure 5: Qualitative comparison between our DISCO (w/ or w/o HAP) and DreamPose on **human dance video generation** with the input of a reference image and a sequence of target poses. Note that the reference image and target poses are from the testing split, where the human subjects, backgrounds, poses are not available during the model training. Best viewed when zoomed-in.

concatenate every consecutive 16 frames to form a sample, to report FID-VID (Balaji et al., 2019) and FVD (Unterthiner et al., 2018). As shown in Table 1, DISCO without human attribute pre-training (HAP) already significantly outperforms DreamPose by large margins across all metrics. Adding HAP further improves DISCO, reducing FID to ~ 38 and FVD to ~ 280. Not surprisingly, classifier-free guidance gives additional advantages to the generation quality of DISCO. The substantial performance gain against the recent SOTA model DreamPose evidently demonstrates the superiority of DISCO. Furthermore, we additionally collect 250 TikTok-style short videos from the web to enlarge the training split to ~600 videos in total. The performance gain has shown the potential of DISCO to be further scaled-up. As shown in Table 2, further incorporating temporal modeling to DISCO brings huge performance boosts to video synthesis metrics, *e.g.*, improving FID-VID by ~ 30 and FVD by ~ 60.

**Qualitative Comparison**. We qualitatively compare DISCO to DreamPose in Figure 5. DreamPose obviously suffers from inconsistent human attribute and unstable background. Without HAP, DISCO can already reconstruct the coarse-grained appearance of the human subject and maintain a steady background in the generated frames. With HAP, the more fine-grained human attributes (*e.g.*, black long sleeves in the left instance and the vest color in the right instance) can be further improved. It is worth highlighting that our DISCO can generate videos with surprisingly good temporal consistency, even without explicit temporal modeling.

Table 2: Video generation comparisons by adding the Temporal Modeling (TM). We employ HAP and CFG by default.

| Method | FID-VID ↓ | FVD ↓ |
|---|---|---|
| DISCO | 59.90 | 292.80 |
| DISCO (w/. TM) | 34.37 | 254.13 |
| DISCO[†] | 55.17 | 267.75 |
| DISCO[†] (w/. TM) | **29.37** | **229.66** |

Table 3: Ablation on architecture designs without HAP. "ControlNet (fg+bg)" and "Attention (fg+bg)" in the first block denote inserting the control condition of reference image (containing both foreground and background) via a single ControlNet or cross-attention modules. "CLIP Global/Local" means using the global or local CLIP feature to represent the reference foreground. "CLIP Local + VAE" combines VAE features with CLIP Local features. Additional ablation results are included in Appendix C.1.

| Method | FID ↓ | SSIM ↑ | PSNR ↑ | LISPIS ↓ | L1 ↓ | FID-VID ↓ | FVD ↓ |
|---|---|---|---|---|---|---|---|
| DISCO | 61.06 | **0.631** | **28.78** | 0.317 | **4.46E-04** | **73.29** | **366.39** |
| *Ablation on control mechanism w/ reference image* (DISCO setting: ControlNet (bg) + Attention (fg) ) | | | | | | | |
| ControlNet (fg+bg) | 65.14 | 0.600 | 28.57 | 0.355 | 4.83E-04 | 74.19 | 427.49 |
| Attention (fg+bg) | 80.50 | 0.474 | 28.01 | 0.485 | 7.50E-04 | 80.49 | 551.51 |
| *Ablation on reference foreground encoding* (DISCO setting: CLIP Local) | | | | | | | |
| CLIP Global | 63.92 | 0.621 | 28.61 | **0.311** | 5.00E-04 | 73.33 | 391.41 |
| CLIP Local + VAE | **59.74** | 0.623 | 28.52 | 0.331 | 4.79E-04 | 77.86 | 406.16 |

Table 4: Ablation analysis of image data size for human attribute pre-training.

| Pre-train Data | Data Size | FID ↓ | SSIM ↑ | PSNR ↑ | LISPIS ↓ | L1 ↓ | FID-VID ↓ | FVD ↓ |
|---|---|---|---|---|---|---|---|---|
| N/A | 0 | 61.06 | 0.631 | 28.78 | 0.317 | 4.46E-04 | 73.29 | 366.39 |
| TikTok | 90K | 50.68 | 0.648 | 28.81 | 0.309 | 4.27E-04 | 69.68 | 353.35 |
| + COCO | 110K | 48.89 | 0.654 | 28.97 | 0.303 | 4.07E-04 | 62.15 | 326.88 |
| + SSHQ | 184K | 44.13 | 0.655 | 29.00 | 0.300 | 3.93E-04 | 64.47 | 325.40 |
| + DpFashion2 + LAION | 700K | **38.19** | **0.663** | **29.33** | **0.291** | **3.69E-04** | **61.88** | **286.91** |

## 4.4 ABLATION STUDY

**Architecture Design**. Table 3 quantitatively analyzes the impact of different architecture designs in DISCO. First, to ablate the control with reference image, we observe that either ControlNet or the cross-attention module struggle haodto handle the control of the whole reference image without disentangling the foreground from the background, leads to inferior quantitative results on most metrics. Though ControlNet-only baseline achieves better FVD scores, our visualizations in Figure 6 show that such architecture still struggles to maintain the consistency of human attributes and the stability of the background. For the encoding of reference foreground, DISCO with CLIP local feature produces better results than the one with CLIP global feature on 6 out of 7 metrics. We also explore to complement the CLIP local feature with VAE feature with a learnable adaptor following DreamPose (Karras et al., 2023), which leads to a slight better FID score but worse performance on all other metrics.

**Pre-training Data Size**. Table 4 investigates the effect of the data size in HAP stage by incrementally augmenting the pre-training data from open-source human image datasets. It is evident that a larger and more diverse pre-training data can yield better downstream results for referring human dance generation. Moreover, compared with "without pre-training" (1st row), adopting HAP on the same TikTok dataset as a self-supervised learning schema (2nd row) can al-

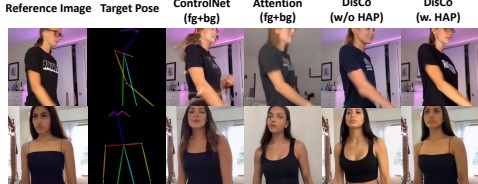

Figure 6: The qualitative comparison between different architecture designs.

ready bring out significant performance gains. The final success of HAP comes from two-sides: 1) learning diverse human attributes from large-scale human image data; 2) the "easy-to-hard" training schema, with HAP focusing on reconstructing human images without pose editing, then learning pose control and implicit appearance distortions brought by motion in the fine-tuning stage.

## 5 CONCLUSION

We revisit human dance synthesis for the more practical social media scenario and emphasize two key properties, *generalizability* and *compositionality*. To tackle this problem, we propose DISCO, equipped with a novel architecture for disentangled control and an effective human attribute pre-training task. Extensive qualitative and quantitative results demonstrate the effectiveness of DISCO, which we believe is a step closer towards real-world applications for user-specific short video content generation. The limitation of DISCO sheds light on potential future directions: (1) incorporating hand keypoints for more fine-grained control of hand pose; (2) extending to more complicated scenarios, such as multi-person dance generation and human-object interaction.

**Repreducibility Statement.** We have clarified enough training and testing details including hyper-parameters, training pipelines, settings in Section 4.1 and Appendix B. All the baseline methods and datasets used in this paper are open-sourced and can be accessed online. Moreover, we include the code in supplementary materials.

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

# Appendix - DISCO: Disentangled Control for Human Dance Generation

This appendix is organized as follows:

- Section A includes comprehensive analysis and comparison between our proposed DISCO and related works.
- Section B demonstrates how DISCO can be readily combined with subject-specific fine-tuning.
- Section C provides more qualitative and quantitative results to supplement the main paper.

## A   DETAILED DISCUSSION ON RELATED WORK

We include additional discussions with the related visual-controllable image/video generation methods, especially the more recent diffusion-based models, due to the space limitation of the main paper. To fully (or partly) maintain the visual contents given a reference image/video, existing diffusion-based synthesis methods can be broadly divided into the following two categories based on their immediate applications:

**Image/Video Editing Conditioned on Text**. The most common approach for preserving specific image information is to edit existing images (Meng et al., 2021; Hertz et al., 2022; Brooks et al., 2022; Kawar et al., 2022; Tumanyan et al., 2022) and videos (Wu et al., 2022; Liu et al., 2023a; Qi et al., 2023; Shin et al., 2023) with text, instead of unconditioned generation solely reliant on text descriptions. For example, Prompt-to-Prompt (Hertz et al., 2022) control the spatial layout and geometry of the generated image by modifying the cross-attention maps of the source image. SDEdit (Meng et al., 2021) corrupts the images by adding noise and then denoises it for editing. DiffEdit Couairon et al. (2022) first automatically generates the mask highlighting regions to be edited by probing a diffusion model conditioned on different text prompts, then generates edited image guided by the mask. Another line of work requires parameter fine-tuning with user-provided image(s). For example, UniTune (Valevski et al., 2022) tries to fine-tune the large T2I diffusion model on a single image-text pair to encode the semantics into a rare token. The editing image is generated by conditioning on a text prompt containing such rare token. Similarly, Imagic (Kawar et al., 2022) optimizes the text embedding to reconstruct reference image and then interpolate such embedding for image editing.

For video editing, in addition to the Follow-your-pose (Ma et al., 2023) and Text2Video-Zero (Khachatryan et al., 2023) discussed in the main text, Tune-A-Video (Wu et al., 2022) fine-tunes the SD on a single video to transfer the motion to generate the new video with text-guided subject attributes. Video-P2P (Liu et al., 2023a) and FateZero (Qi et al., 2023) extend the image-based Prompt-to-Prompt to video data by decoupling the video editing into image inversion and attention map revision. However, all these methods are constrained, especially when the editing of the content cannot be accurately described by the text condition. We notice that a very-recent work Make-A-Protagonist (Zhao et al., 2023) tries to introduce visual clue into video editing to mitigate this issue. However, this approach, while innovative, is still met with limitations. On the one hand, it still struggles to fully retain the fine-grained human appearance

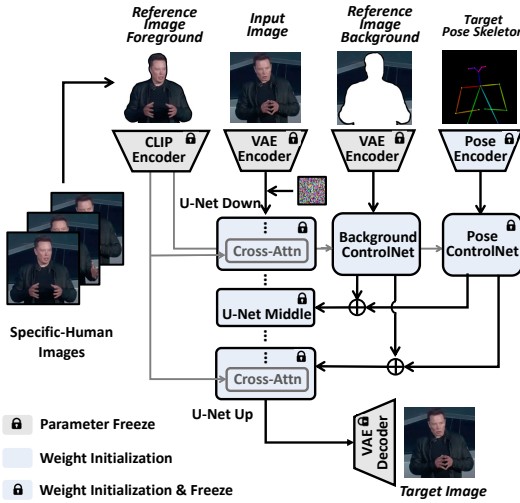

Figure 7: The model architecture for further subject-specific fine-tuning.

and background details; on the other hand, it still requires a specific source video for sample-wise fine-tuning which is labor-intensive and time-consuming. In contrast, our DISCO not only readily facilitates human dance synthesis given any human image, but also significantly improves the

faithfulness and compositionality of the synthesis. Furthermore, DISCO can also be regarded as a powerful pre-trained human video synthesis baseline which can be further integrated with various subject-specific fine-tuning techniques (see section B for more details).

**Visual Content Variation**. For preserving the visual prior, another line of work (Justin & Lambda, 2022; Ramesh et al., 2022; Esser et al., 2023) directly feeds the CLIP image embedding into the diffusion model to achieve image/video variation. However, these approaches struggle to accurately control the degree as well as the area of the variation. To partially mitigate this problem, Dream-Booth (Ruiz et al., 2022) and DreamMix (Molad et al., 2023) necessitate multiple images to fine-tune the T2I and T2V models for learning and maintaining a specific visual concept. However, the precise visual manipulation is still missing. In this paper, we propose a disentangled architecture to accurately and fully control the human attribute, background and pose for referring human dance generation.

## B  SUBJECT-SPECIFIC FINETUNING

As mentioned in the main paper, our DISCO can be flexibly integrated with existing efficient fine-tuning techniques for even more fine-grained human dance synthesis. This is particularly beneficial when facing out-of-domain reference images, which appear visually different to the TikTok style images. Figure 7 presents the framework for subject-specific fine-tuning, which is easily adapted from the framework presented in the main text (Figure 2a). Rather than utilizing a set of videos of different human subjects for training, subject-specific fine-tuning aims to leverage limited video frames of a specific human subject (*e.g.*, the video of Elon Mask talking about Tesla Model 3 in Figure 7 or even anime in Figure 9) for better dance synthesis. Compared to the standard fine-tuning, we additionally freeze the pose ControlNet branch and most parts of U-Net to avoid over-fitting to the limited poses in the subject-specific training video, only making the background ControlNet branch and the cross-attention layers in U-Net trainable. We also explored the widely-used LoRA (Hu et al., 2021) for parameter-efficient fine-tuning and observed similar generation results. As this is not the main focus of this paper, we leave other advanced techniques to future explorations along this direction.

**Implementation Details**. The model weights are initialized with the model fine-tuned on the general TikTok dancing videos. We train the model on 2 NVIDIA V100 GPUs for 500 iterations with learning rate $1e^{-3}$, image size $256 \times 256$ and batch size 64. The randomized crop is adopted to avoid over-fitting. The subject-specific training videos range from 3s to 10s, with relatively simple poses.

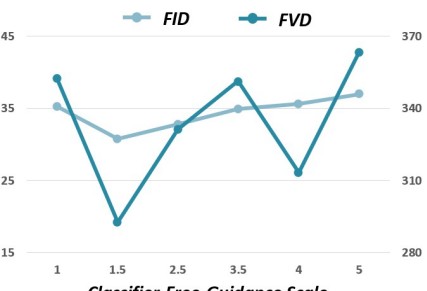

Figure 8: The effect of different classifier-free-guidance scale.

**Qualitative Results**. We test the subject-specific fine-tuning on various out-of-domain human subjects, including real-world celebrities and anime characters. After training, we perform the novel video synthesis with an out-of-domain reference image and a random dance pose sequence sampled in TikTok training set. As shown in Figure 9, upon additional fine-tuning, DISCO is able to generate dance videos preserving faithful human attribute and consistent background across an extensive range of poses. This indicates the considerable potential of DISCO to serve as a powerful pre-trained checkpoint.

## C  ADDITION RESULTS

### C.1  QUANTITATIVE RESULTS

We show the full ablation results on architecture design in Table 5. In what follows, we focus on discussing results that are not present in the main text. In the first block of the table, we copy over the results from the full instances of DISCO, with or without HAP on TikTok Dance Dataset for reference.

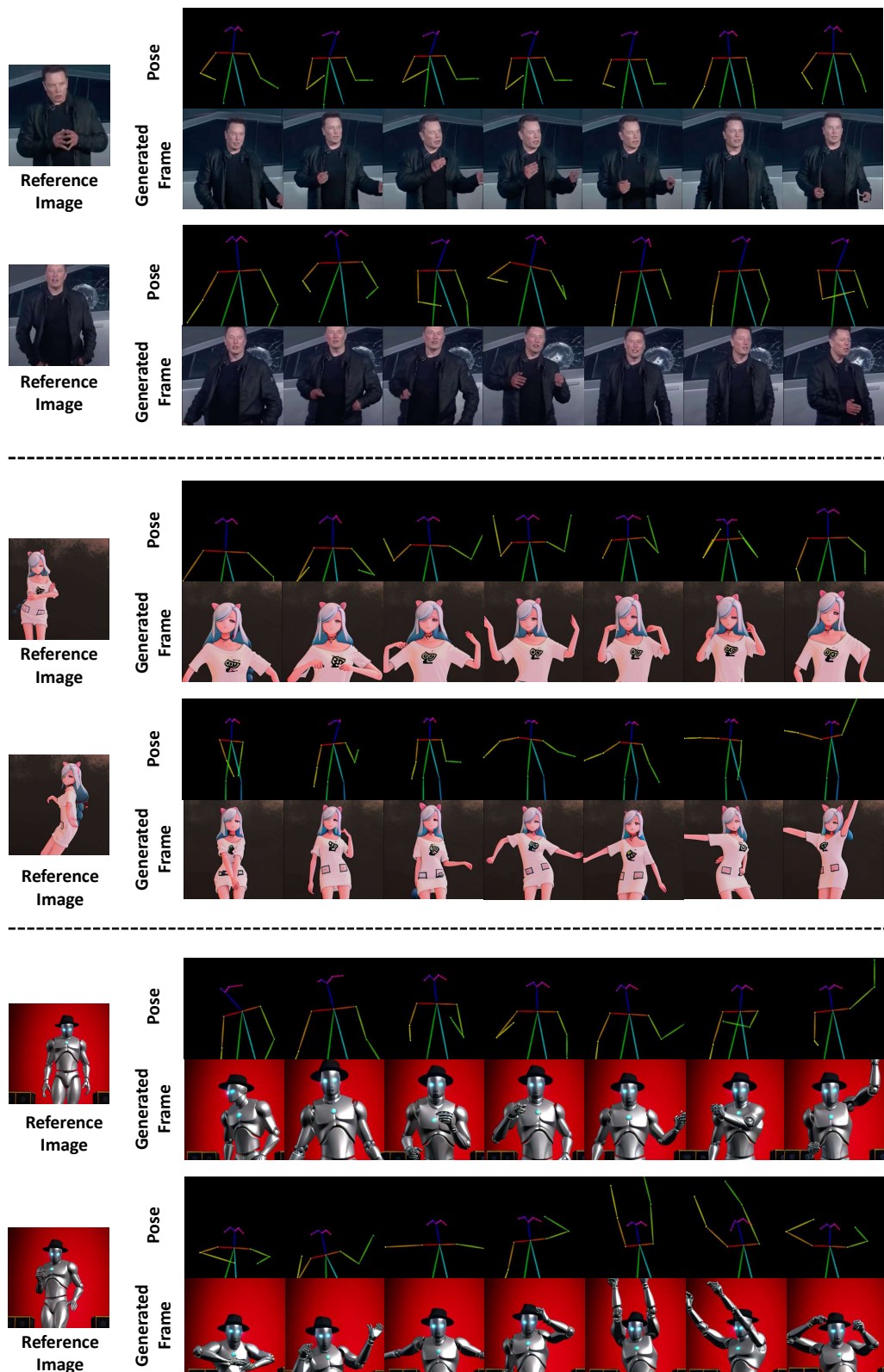

Figure 9: The synthesis frames for out-of-domain human subject after subject-specific fine-tuning guided by the pose sequence extracted from the TikTok dataset.

Table 5: Additional ablation results on architecture designs. "ControlNet (fg+bg)" and "Attention (fg+bg)" in the second block denote inserting the control condition of reference image (containing both foreground and background) via a single ControlNet or cross-attention modules. "HAP w/ pose" denotes adding pose ControlNet path with pose annotation into HAP. "ControlNet-Pose" Init. means initializing pose ControlNet of fine-tuning stage with the pre-trained ControlNet-Pose (Zhang & Agrawala, 2023) checkpoint.

| Method | FID ↓ | SSIM ↑ | PSNR ↑ | LISPIS ↓ | L1 ↓ | FID-VID ↓ | FVD ↓ |
|---|---|---|---|---|---|---|---|
| DISCO | 61.06 | 0.631 | 28.78 | 0.317 | 4.46E-04 | 73.29 | 366.39 |
| DISCO + TikTok HAP | 50.68 | 0.648 | 28.81 | 0.309 | 4.27E-04 | 69.68 | 353.35 |
| *Ablation on control mechanism w/ reference image* (DISCO setting: ControlNet (bg) + Attention (fg) ) | | | | | | | |
| ControlNet (fg+bg, no SD-VAE) | 83.53 | 0.575 | 28.37 | 0.411 | 5.35E-04 | 89.13 | 551.62 |
| ControlNet (fg+bg) | 65.14 | 0.600 | 28.57 | 0.355 | 4.83E-04 | 74.19 | 427.49 |
| Attention (fg+bg) | 80.50 | 0.474 | 28.01 | 0.485 | 7.50E-04 | 80.49 | 551.51 |
| *Ablation on HAP w/ pose* (DISCO setting: HAP w/o pose) | | | | | | | |
| TikTok HAP w/ pose | 51.84 | 0.650 | 28.89 | 0.307 | 4.16E-04 | 68.55 | 346.10 |
| *Ablation on initializing w/ pre-trained ControlNet-Pose* (DISCO setting: initialize with U-Net weights) | | | | | | | |
| ControlNet-Pose Init. | 62.18 | 0.633 | 28.37 | 0.320 | 4.46E-04 | 72.98 | 389.47 |
| ControlNet-Pose Init.+TikTok HAP | 55.81 | 0.641 | 28.69 | 0.316 | 4.43E-04 | 78.13 | 363.38 |

As mentioned in the main text, we propose to use the pre-trained VQ-VAE from SD, instead of four randomly initialized convolution layers in the original ControlNet for encoding the background reference image. In the second block of the table, we ablate this design by comparing two models, (1) "ControlNet (fg+bg)", inserting the control condition of reference image (containing both foreground and background) via a single ControlNet, with VQ-VAE encoding and (2) "ControlNet (fg+bg, no SD-VAE)", inserting the control condition of reference image via a single ControlNet with four randomly initialized convolution layers as the condition encoder. We note that the pre-trained VQ-VAE can produce a more descriptive dense representation of the reference image, contributing to better synthesis results (FID 65.14 v.s 83.59).

In the third block of the table, we investigate whether adding pose condition into human attribute pretraining is beneficial to the downstream performance. We observe that integrating pose into HAP leads in similar results, but requires additional annotation efforts on pose estimation.

Last but not least, we examine on the initialization of the pose ControlNet branch. Specifically, we try to initialize from the pre-trained ControlNet-Pose checkpoint (Zhang & Agrawala, 2023) during fine-tuning. The results are shown in the last block of Table 5. Without HAP, the performance is comparable to DISCO, but it gets much worse than DISCO when both are pre-trained with HAP. This is because that the ControlNet-Pose is pre-trained with text condition and can not fully accommodate referring human dance generation with the reference image condition. After HAP, such gap is further enlarged, leading to even worse results. In Figure 8, we show the effect of varying the classifier-free-guidance scale. We can find that scale of 1.5 gives the best quantitative results for both image-wise and video-wise fidelity.

## C.2 QUALITATIVE RESULTS

DISCO can be easily adapted to different image size. For example, we show more qualitative results of human image editing in Figure 10 with image size of $256 \times 384$ to include more human body. Please note that most videos of the TikTok dataset are relatively close to the camera. However, we can see that DISCO can handle both partial and full human body synthesis even with large changes in viewpoints and rotations in the human skeleton. More results for video generation are shown in Figure 12.

Figure 11 compares the video synthesis results with baseline architectures, to supplement Figure 6 of the main text. With a sequence of continuous poses, we can discern more clearly that both ControlNet-only and Attention-only baseline fail to maintain the consistency of human attributes and background, leading to less visually appealing generations than our DISCO.

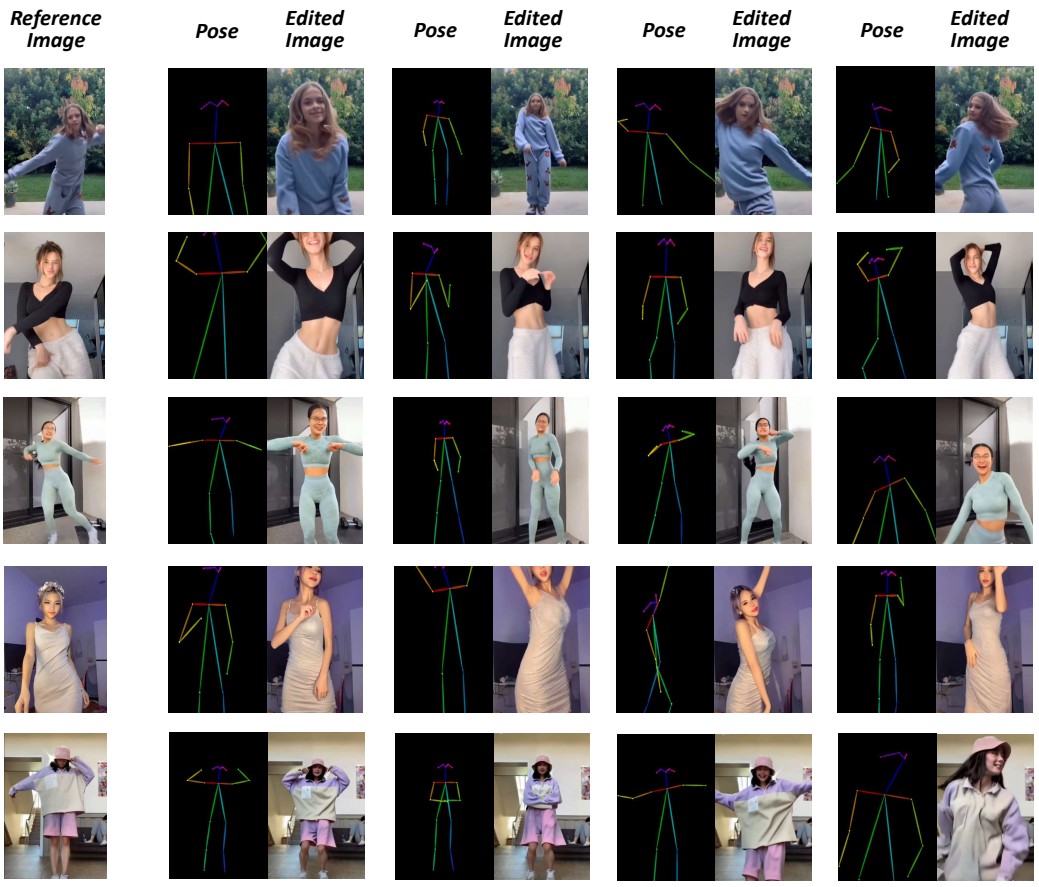

Figure 10: More visualizations for different image ratios (*i.e.*, vertical image) and various human views (*e.g.*, from close-view to full-body) of human image editing.

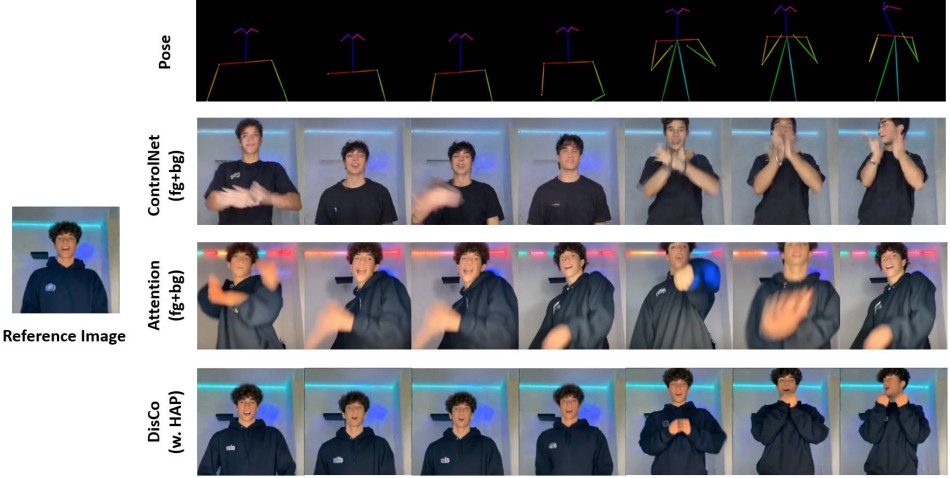

Figure 11: The qualitative comparison between different architecture designs for the video frame generation.

**Pose Sequence**

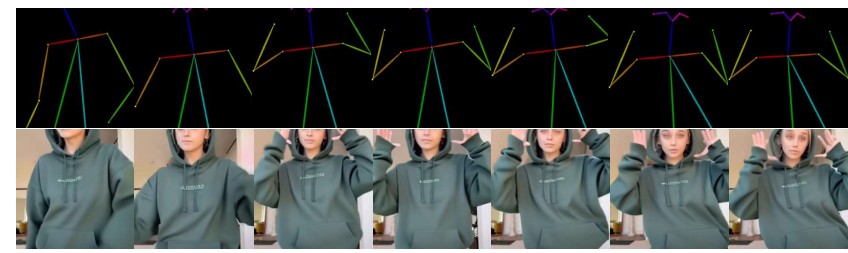

**Reference
Image**

**Generated Frame**

**Pose Sequence**

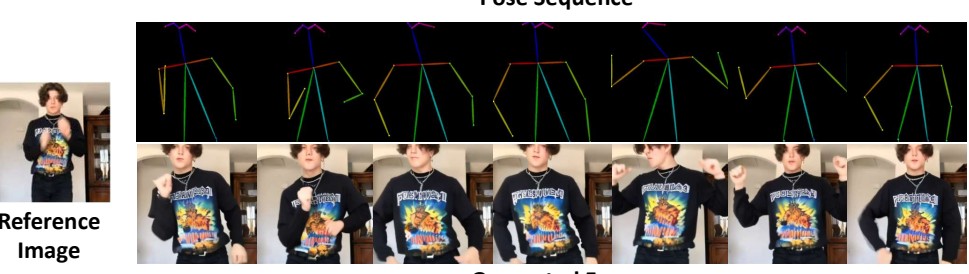

**Reference
Image**

**Generated Frame**

**Pose Sequence**

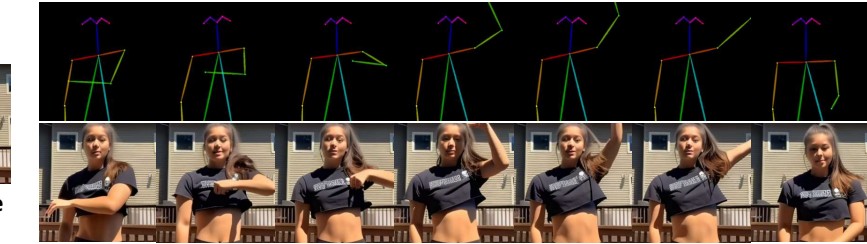

**Reference
Image**

**Generated Frame**

**Pose Sequence**

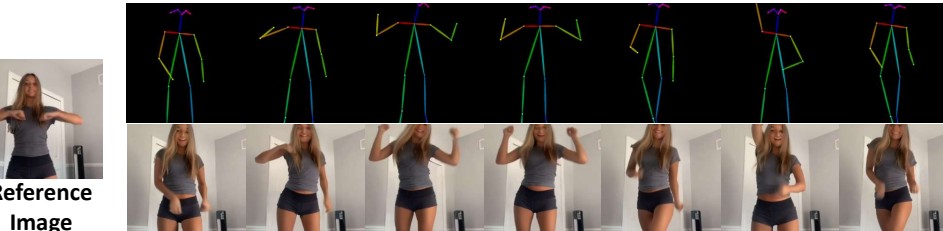

**Reference
Image**

**Generated Frame**

**Pose Sequence**

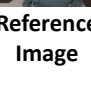

**Reference
Image**

**Generated Frame**

Figure 12: More qualitative examples for video generation.

