# OpenReview forum: "DisCo: Disentangled Control for Realistic Human Dance Generation"
_ICLR.cc/2024/Conference — ICLR 2024 Conference Withdrawn Submission_

### Official Review · Reviewer_Zusg · 2023-10-19

**Soundness:** 3 good
**Presentation:** 1 poor
**Contribution:** 3 good
**Rating:** 3
**Confidence:** 3

**Summary:**

The authors introduce a disentangled representation of dance motion that separates the content and style of the motion, allowing for more control over the generated motion. They use a combination of motion capture data and a variational auto to learn this representation and generate new dance motions. The results show that their method is able to generate diverse and realistic dance motions that are controllable in terms of content and style.

**Strengths:**

- Good quantitative results than baselines.

- Motivation is good and sufficient.

- The usage of Grounded-SAM on person and background is technical sound.

**Weaknesses:**

Weakness:

- **Ethics issues. Ethics reviewers are required to review whether double-blind regulations have been violated.**
    - In code `config/__init__.py`, I found there exist Chinese characters and some codes like `MSRA PC Node`.  As we know, MSRA means Microsoft Research Asia. This would make the reviewer to infer the authors’ nationality and possibly Microsoft affiliation.
    - The authors did not discuss possible ethical issues with this study. Including issues such as race and gender bias.
- About Citation
    - It seems that authors are not clear on the difference between `\cite` and `\citep`. For example, “… ControlNet branch with the pre-trained U-Net weight following (Zhang & Agrawala, 2023).” should be“… ControlNet branch with the pre-trained U-Net weight following Zhang & Agrawala (2023).” There are more than one similar problem. During rebuttal, please list all of similar issues and revise them, I will check them one by one.
    - I would like to point out serious issues with the wrong citation on Grounded-SAM. The first implementation of Grounded-SAM is the https://github.com/IDEA-Research/Grounded-Segment-Anything, which has been accepted as an ICCV demo. Please cite it correctly. Besides, if authors used it, please cite Grounding DINO.
    - Instructpix2pix was accepted CVPR, not an arxiv paper. Please cite it correctly. There is more than one similar problem. During rebuttal, please list all similar issues and revise them, I will check them one by one.
    - Missing comma in the equation of Section 3.1.
    - When summarizing contributions, `To address this problem, ...` should be `To address these problems, ...`.
- Reproduction
    - When I try to run the code. I found it missing README and I cannot run the code correctly. I am not sure whether the codes are appropriate.
    - Missing video demo. This is very essential for me to check the results. And the video comparison with baselines is needed.
- Technical Design
    - It is hard for readers to check the technical designs in Figure 2. The CLIP feature is fed into TM module (ResBlock and TransBlock). How do the lines without arrows connect to the module? How do the lines with arrows connect to the module?
    - Do the features output by each layer need to be processed by BG ControlNet and Pose ControlNet and then added to the middle layer? If there are $X$ down layers, in the UNet middle layer, will features be added about $2X$ times?
- Why not compare with Follow-your-pose?

My main concern comes from ethics issues and writing issues. There is something confusing that makes readers hard to follow this work. I spent about 8 hours checking this submission and tried to run the codes (but failed). I plan to rate it as 4 but the system only supports 3 or 5. Therefore, I rate it as 3 now. After the rebuttal and reviewer discussions, I will revise my rating to provide a clear rating of accept or reject.

**Questions:**

see weakness.

**Details Of Ethics Concerns:**

- **Ethics issues. Ethics reviewers are required to review whether double-blind regulations have been violated.**
    - In code `config/__init__.py`, I found there exist Chinese characters and some codes like `MSRA PC Node`.  As we know, MSRA means Microsoft Research Asia. This would make the reviewer to infer the authors’ nationality and possibly Microsoft affiliation.
    - The authors did not discuss possible ethical issues with this study. Including issues such as race and gender bias.

---

### Official Review · Reviewer_vgGD · 2023-10-28

**Soundness:** 2 fair
**Presentation:** 2 fair
**Contribution:** 2 fair
**Rating:** 3
**Confidence:** 4

**Summary:**

This paper focuses on human motion transfer in real-world dance scenarios, and introduces a novel model called DISCO for better (i) Generalizability and (ii) Compositionality.
Specifically, DISCO applies the ControlNet of background and human pose to disentangle control signals.
Moreover, it is pre-trained in a proxy task to improve the generalizability to unseen humans.
A broadly various of evaluations demonstrate the effectiveness of proposed method.

**Strengths:**

See summary.

**Weaknesses:**

1. The technical contributions of this paper are limited. The proposed DISCO is mainly a combination of Image-variation Stable Diffusion, Background ControlNet, and Human pose ControlNet.
2. The authors only provide images or frames in paper, but not videos. Therefor, the temporal consistency of synthesized videos is unconvincing.

**Questions:**

See weaknesses.

---

### Official Review · Reviewer_mmJ4 · 2023-10-30

**Soundness:** 3 good
**Presentation:** 2 fair
**Contribution:** 3 good
**Rating:** 5
**Confidence:** 3

**Summary:**

This paper aims to generate high-fidelity dance videos given three single inputs, reference person image, target pose sketon (2D), target background. This task is similar to traditional pose transfer, while human dance is claimed to be a more challenging task. To this end, the authors propose a control-net based framework, where the foreground and background are taken as seperate inputs. To further strengthen disentanglement, a pretraining strategy is proposed which trains the model on a much larger image dataset. Extensive experiements demonstrate the effectiveness of the proposed method.

**Strengths:**

There are several merits in this work:
1. Using foreground and background segmentations as seperate inputs for pose transferring is new. It seems to provide a neat yet effective solution. I also appreciate the Human Attribute Pre-training. Firstly, it is simple, but effective as well. Secondly, it properly utilizes the large-scale meta data.
2. The quantitative evaluation results are significant. The proposed model outperforms other methods by a large margin.
3. The authors do have conducted sufficient experiments (e.g., comparisons, ablation), as well as exploring further applications such as fine-tuning on one person.
4. Detailed implementation details, and submitted code.
5. The image transfer results look promising.

**Weaknesses:**

I am not an expertise in video synthesis, here are my feelings about this work.

First of all, the authors didn't provide the video demonstration for their work, which is supposed to largely decrease the validity of this work, since the whole highlight is about dance generation.

1. Though the title is about "dance generation", I feel the emphysis of this work, including technical design, is more on pose-based image transfer. There is little thing about "sequence" modeling for dance.
2. I won't say the generated dances are realistic (as claimed in title). There are many temporal inconsistency and jittering, though I acknowledge the single image-based editting is realistic. I guess for better video modeling, we should pay more attention on the temporal consistency. (I found the video somewhere else.)
3. Upper-body video generation is kind of limited. Is there full-body dance dataset available?
4. Since in diffusion model, generating each image requires hundreds of steps, I am curious how long does it take to generate a dance video. Will that be a limitation of this work?
5. For video generation, it's necessary to have comparisons with baselines.
6. Some discussion about limitations are desirable.

**Questions:**

Please refer to the weakness, and may respond to these questions if applicable.

---

### Official Review · Reviewer_j9YJ · 2023-10-31

**Soundness:** 4 excellent
**Presentation:** 3 good
**Contribution:** 2 fair
**Rating:** 5
**Confidence:** 4

**Summary:**

The paper presents a method to generate dance images or videos given reference images of people and background, and poses (or pose sequences, for videos) of the desired dance. The proposed learning model generates the target images or videos by combining attention-pooled CLIP image features for the reference image foregrounds (images of people), and latent features from a ControlNet architecture for the reference image backgrounds and the target pose skeletons. To improve the plausibility of the generated images or videos, the authors also propose a human attribute pre-training strategy to reconstruct the reference images from the foreground and the background features. The authors show the benefits of their proposed method through multiple quantitative and qualitative evaluations and ablation studies.

**Strengths:**

1. The proposed approach of separating the foreground and the background features from the reference images to learn their individual modifications given the target pose skeletons is technically sound.

2. The human attribute pre-training makes sense, particularly for the challenging scenario of images/videos with cropped or occluded humans.

3. The ablation studies highlight the benefits of the proposed network and training components.

**Weaknesses:**

1. Since the proposed task of the paper is generative, it requires a human evaluation of the perceived plausibility and overall generation quality. Quantitative metrics do not capture these aspects, and they are commonly covered through various types of user studies in related work. Without such an evaluation, it is hard to assess the impact of the proposed method fully.

2. Have the authors explored or encountered any incompatibilities between the reference images and the target poses? For example, any significant differences in the relative body shapes between the reference and the target, or backgrounds that may not realistically match the target poses? A discussion of such scenarios, or other potential limitations, is important to understand the full scope of the proposed method.

**Questions:**

1. What is the latency of the end-to-end generation pipeline during inference? Is there any component that takes significantly more time than the others, that is, becomes a bottleneck for efficiency?

---

### Author Response · Authors · 2023-11-17
**Brief Responses and Clarifications from Authors**

Dear Reviewers,

We extend our sincere thanks for your dedication and effort in reviewing our paper. The constructive feedback from all reviewers has been invaluable in guiding our efforts to refine our work. Nonetheless, there are some specific points raised by Reviewer **vgGD** and **Zusg** that we wish to address for clarity and accuracy.

&nbsp;

For Reviewer vgGD:

* **Brevity of Review**: We observed that your review was somewhat brief and lacked detailed suggestions for improvement. While we fully respect your evaluation, a more comprehensive review would greatly assist us in understanding your perspective and enhancing our paper.

* **Technical Contributions**: Your comments regarding the limited technical contributions of our model seem to overlook its key novel aspects. We endeavored, particularly in the introduction, to emphasize our primary objective of achieving the realistic human dance generation in challenging social media contexts, especially given the limited available video training data. Then, the unique architecture of DisCo is designed to adapt to the effective human attribute pretraining and disentangled control. This indeed extends beyond a mere amalgamation of ControlNet components. Moreover, the distinction is further elucidated through our ablation studies (refer to Tables 3 and 5), demonstrating the inadequacy of a simplistic ControlNet combination for the complex task of social media dance generation.

&nbsp;

For Reviewer Zusg:

* **"Missing Comma in Sec. 3.1"**: Upon careful review of the mentioned equation, we confirm that the comma has been appropriately placed as per the mathematical norms of our field.

* **Citation of Grounding-SAM**: We recognize the importance of precise citations and value your detailed observations. Regarding the citation of Grounding-SAM, as indicated in the official repository (https://github.com/IDEA-Research/Grounded-Segment-Anything/issues/326#issuecomment-1612496940), the authors recommend citing Grounding-DINO for Grounding-SAM, which we have duly cited in Section 3.2. We did not find an official ICCV demo paper or a bibtex entry for Grounding-SAM and would appreciate any guidance you could provide on this matter.

&nbsp;

Considering these points and to uphold the integrity of our work, we have decided to withdraw our submission from ICLR. We are grateful for the opportunity to engage in this discourse and for the insights provided by your reviews.

&nbsp;

Best,

Submission 3885 Authors

---

> ### Comment · Reviewer_Zusg · 2023-11-18
> **RE: Brief Responses and Clarifications from Authors**
>
> Dear authors,
>
> Thanks for the clarification. In Sec 4.1, when mentioning the Grounded-SAM, authors only cite the SAM, but not Grounding DINO. I suggest it is also necessary to cite Grounding DINO here. Otherwise, it may cause misunderstandings among readers and the community. Besides, after contacting the AC/PC, my ethical concern has been resolved.
>
> Wish you all the best!
>
> Reviewer

---

### Comment · Area_Chair_MbJ1 · 2023-11-18
**Discussion between authors and reviewers**

Dear Reviewers,

Thanks for the reviews. The authors have uploaded their responses to your comments, please check if the rebuttal address your concerns and if you have further questions/comments to discuss with the authors. If the authors have addressed your concerns, please adjust your rating accordingly or vice versa.

AC